# Towards the Validation of an Observational Tool to Detect Impaired Drivers—An Online Video Study

**DOI:** 10.3390/ijerph19127548

**Published:** 2022-06-20

**Authors:** Tanja Baertsch, Marino Menozzi, Signe Maria Ghelfi

**Affiliations:** 1Department of Health Sciences and Technology, ETH Zürich, 8092 Zurich, Switzerland; mmenozzi@ethz.ch; 2Zurich State Police, Airport Division–Research and Development, Zurich-Airport, 8058 Kloten, Switzerland; signe@gmx.ch

**Keywords:** accident prevention, workplace drug testing, impaired driving, video-based training

## Abstract

Abuse of alcohol and other drugs is a major risk factor at work. To reduce this risk, workplace drug testing is performed in transportation and other industries. VERIFY, an observational method, is one of the key elements in a procedure adopted by the police of the canton of Zurich, Switzerland, for detecting impaired drivers. The observational method has been successfully applied by adequately trained police officers since 2014. The aim of this study is to examine the interrater reliability of the observational method, the effect of training in use of the method, and the role of having experience in the police force and traffic police force on the outcome when rating a driver’s impairment. For this purpose, driver impairment in staged road traffic controls presented in videos was rated by laypeople (*n* = 81), and police officers without (*n* = 146) and with training (*n* = 172) in the VERIFY procedure. In general, the results recorded for police officers with training revealed a moderate to very good interrater reliability of the observational method. Among the three groups, impaired drivers were best identified by officers with training (ranging between 82.6% and 89.5% correct identification). Trained officers reported a higher impairment severity of the impaired drivers than the other two groups, indicating that training increases sensitivity to signs of impairment. Our findings also suggest that online video technology could be helpful in identifying impaired drivers. Trained police officers could be connected to a road traffic control to make observations via live video. By this method efficiency and reliability in detecting abuse of alcohol and other drugs could be improved. Our findings also apply to workplace drug testing in general.

## 1. Introduction

Across cultures, alcohol, other psychoactive substances, medicinal drugs, sleepiness, and their combined effects are serious issues that impact occupational safety [1,2,3,4,5,6,7,8] and lead to economic costs, expenditures on health care, and costs related to law enforcements [9,10]. In particular, substance abuse and sleepiness may lead to accidents, adversely affecting people and environments, such as in transportation (aviation, train, and road traffic in general) and nuclear power plants. To keep work safe, workplace drug testing (WDT) has been introduced in various industries [11,12], for example, the aviation sector [13], to identify drug abuse at an individual level. Similarly, objective roadside measurements, such as breathalyzers for estimating blood alcohol content or oral drug screening devices, are used in police traffic controls to foster traffic safety. Because of their high specificity to particular drugs, roadside drug tests are blind to other drugs, such as new psychoactive substances, most medicinal drugs, and combinations of substances [14,15,16].

In addition to objective measurements, the so-called impairment tests are applied as a subjective means to detect impairment by evaluating the behavior and appearance of the examinee. With subjective methods, effects on examinees’ behavior and appearance are recorded and evaluated rather than measuring the cause itself, which is a major advantage of subjective methods over objective approaches. Thus, the issues with screening tests measuring only specific substances can be overcome. However, currently widely used subjective methods are time-consuming. In police traffic control, for instance, the drug recognition expert evaluation used in the United States, which includes an interview, recording clinical signs, and performing psychomotor tests, takes 90 min [17] and the Standardized Field Sobriety Tests, including the gaze nystagmus test, walk and turn, and one-leg stand, takes approximately 10 min [18]. In addition to being time-consuming, impairment tests are biased by subjectivity and depend on expertise in correctly administering them.

Overall, subjective methods can be helpful in complementing WDT and in assessing fitness for work. Considering its successful application since 2014, the subjective observational method called ‘VERIFY’ (a German acronym: *Verfahren zur Identifikation von Fahrunfähigkeit*—procedure for identification of driving impairment) is a promising tool to overcome various problems of the currently widely used strategies of impairment detection. VERIFY was developed by police officers, who are experts in the detection of impaired drivers, with the collaboration of scientists in pharmaceutical science. VERIFY has proven to be highly effective as the results of observational evaluations were in good agreement with the results of blood and urine analyses and/or the opinion of experts of the Institute of Forensic Medicine on the overall case. According to Boll 2019 [19], in 85% of VERIFY cases where impairment was detected, the driving impairment was confirmed by analytical results for the blood and urine samples of the examinees or/and the expert opinions on the overall cases.

VERIFY consists of a checklist in which observations of driver’s appearance and behavioral factors related to ocular functions, cognition, motor functions, and others are systematically reported. The decision of whether a driver is impaired is based on the totality of the evidence and not simply on one sign of impairment. Before applying VERIFY in practice, police officers must complete a training in which they apply VERIFY during road traffic controls under the guidance of experts. A detailed description of the VERIFY method is given in Section A.1, and a sample of the checklist is shown in Figure 1.

The present study aims to investigate the interrater reliability of the checklist (Figure 1) and the agreement of the rating with the state of impairment of a driver. Furthermore, the effect of the training is examined by comparing ratings of driving impairment between police officers who underwent the training and police officers without training. The latter group obtains reasonable suspicion by objective methods and by observation of signs of impairment but without having the possibility to conduct and report observations as systematically as in the VERIFY procedure. A third group consisting of laypeople was included in the comparison as a control. It is hypothesized that trained police officers are more reliable in identifying impaired drivers than police officers without training. Furthermore, as it is widely assumed that experts perceive risks differently from laypeople and that their judgments are more truthful than those made by laypeople [20], it is hypothesized that laypeople perceive driving impairment as less severe than police officers. Finally, the effect of experience in the police force and in the traffic police force on rating outcomes is investigated.

## 2. Materials and Methods

Our study required participants to use the observational tool, VERIFY, for rating the impairment of drivers in a police traffic control. To avoid variations caused by the direct observation of real-road traffic controls, ratings were performed based on recorded videos showing staged controls of impaired and unimpaired drivers. Video observations have been reported to be a reliable substitute for direct observations in a variety of applications, such as observing posture [21], assessing mental health [22] or depression [23], or detecting lies when conducting police interrogations [24].

Due to ethical considerations and to overcome limitations in recruitment, the roles of drivers in the videos showing staged controls were played by unimpaired actors. According to a study by Paquette and Prince [25], drug recognition experts who were trained solely through simulated evaluations with unintoxicated actors are as effective in detecting intoxicated persons in the field as police officers trained conventionally with intoxicated persons. The strategy of using simulated patients (SP) has successfully been used in medical education since the 1950s [25] and was found to have many advantages over using real patients, such as SP are available at any time, the use of SP avoids mistreatment of real patients, and SP are evaluated without embarrassment of the student [26]. Conclusively, there is evidence that simulated impairment detections are an effective alternative to impairment detections with real impaired persons. Video observations do not require observers to be present at a specific place and at a specific time. Instead, they enable them to efficiently collect observational data using a standalone web-based method.

### 2.1. Participants

Three groups were involved in the study: laypeople, police officers without VERIFY education, and VERIFY-trained police officers (i.e., police officers who underwent VERIFY education).

Laypeople were recruited via a participant panel provided by the Department of Psychology at the University of Zurich. Laypeople were selected according to demographical, cultural, and professional criteria. They had to be 18 years or older, possess a driving license, understand Swiss German, and never have worked in the police force before. Police officers with and without VERIFY education were recruited by the traffic police of the canton of Zurich following the same criteria regarding age, driving license, and language as used in laypeople. In total, 151 police officers without VERIFY education, 173 VERIFY-trained police officers, and 82 laypeople were recruited.

To improve the quality of the collected data, a seriousness check was performed retrospectively [27]. As the total duration of all videos was 15 min, participants who took less than 15 min to complete the study were excluded because they did not watch all videos to the end. In total, seven participants were excluded from the analyses. For the three groups, the sociodemographics of the included participants and the duration for completing the ratings are presented in Table 1.

The self-reported experience levels of police officers in traffic police or in other police forces are presented in Table 2.

### 2.2. Material

Six videos of staged road traffic controls, each lasting approximately two and a half minutes, were recorded during a VERIFY training session in which police officers practiced the VERIFY procedure. The videos demonstrated two police officers stopping a driver, conversing, and examining their pupils. To simulate pupils of impaired drivers, medical educational videos displaying the effects of drugs on pupils were edited into the videos. The size of the edited pupils was the same for the left and right eyes in all videos. The role of all actors in the videos were played by real police officers, and the officer playing the impaired driver pretended to be impaired. There were three scenes with unimpaired drivers and three scenes with impaired drivers. A short description of the content of each video is presented in Table 3.

### 2.3. Procedure

The online study was programmed using Unipark survey software. A link to the online experiment was sent to the participants via e-mail. The experiment could be conducted on a computer at any place or time.

Prior to the experiment, participants were provided with an introductory text explaining the aim of the study, which is to investigate how the assessment of a driver’s ability to drive differs between police officers and laypeople. The text also explained that the task was to watch six videos of staged road traffic controls, rate the driver’s impairment severity, and state the reasons for the suspicion if driving impairment was suspected.

Participants were asked to conduct the experiment alone and without interruption. Before the experiment started, participants answered sociodemographic questions (gender, age). In addition, laypeople were asked for their last educational achievement whereas police officers were asked for years of experience as a police officer and as a road traffic police officer. The data was collected anonymously. A numerical identifier was associated to the data collected with the questionnaire. After that, participants watched a short trial video to check the video and audio quality of their computer. Subsequently, the six videos were presented in a randomized order. Participants were instructed to watch all six videos in a row without repetition or interruption. From the recorded data, it was checked whether the videos were watched only once and without interruption.

After each video, participants in all groups were first asked whether they thought the driver in the video was unimpaired (dependent variable I). This question was answered with a yes or no. Subsequently, participants were asked to indicate the extent to which they agreed or disagreed with the following statements:(I)The driver complied with the instructions of the police.(II)The driver showed conspicuous behavior.(III)The driver showed abnormalities in movement.

A five-point Likert scale ranging from “strongly disagree” to “strongly agree” was used to report the extent of agreement with the statements (II)–(IV).

In cases where the driver was classified as impaired (variable I), participants belonging to the group without VERIFY education or laypeople were further asked to state at least one reason for their suspicion. The group with VERIFY-trained police officers reported their observations by completing the VERIFY checklist (Figure 1). The checklist consists of 49 categorical items (e.g., slurred speech, slow reaction, the smell of alcohol, etc.), two interval-scaled items (right and left pupil sizes), and three open-ended questions. The checklist includes a pupil template with different pupil sizes in mm to help estimate pupil size. Participants were informed that the items about the odor and the lightning condition at the control location should not be considered, as these could not be evaluated in the staged videos.

The study protocol was approved by the ethical committee of ETH Zürich (approval No. 2021-N-03).

### 2.4. Data Analysis

The data were analyzed using SPSS version 26.0 (IBM Corp., Armonk, NY, USA, 2019). Pearson’s χ^2^ was calculated to examine the variation of the assessment of impairment (dependent variable 1) across groups. For significance (*p* ≤ 0.05), the Bonferroni post hoc test was conducted. Cramer’s V was calculated as an effect size measure [28].

On the evaluation results of each video, Fleiss’ kappa [29] was run to investigate the interrater reliability of the VERIFY checklist. The guidelines of Landis and Koch (1977) [30] were used to classify the level of agreement with respect to the value of Fleiss’ kappa. This interrater reliability calculation excluded the open-ended questions and the items about odor and lighting conditions at the control location. As pupil size may vary depending on the types of drugs and the types of drugs influencing the staged impaired driver were not specified in our study, ‘left pupil size’ and ‘right pupil size’ were not used for interrater evaluations. Boxplot distributions of sizes were used as a means for evaluating the accuracy in reporting pupil size.

Cronbach’s alpha was used to determine the internal consistency coefficient of the three statements on driver impairment (see dependent variables II–IV in Section 2.3: Procedure). A score of driving impairment severity was formed based on the three dependent variables II–IV. For this purpose, the coding of the dependent variable II was first reversed, and then the means of agreement with the statements (variables II–IV) were calculated for each video.

To investigate whether the three groups of participants differed in estimating the severity of driving impairment, one-way analyses of variances (ANOVA) with Bonferroni post hoc tests were conducted using the score of impairment recorded for each video. As the homogeneity of variance was only met in results related to videos 1, 2, 3, and 6, Welch ANOVAs were applied to the results of videos 4 and 5.

The evaluation of the reported state of the driver (impaired/not impaired) and the severity of driving impairment included an examination of whether having experience in the police force and in the traffic police force affected the assessment of driving impairment. For this purpose, ANOVA was conducted with the groups of police officers with experience in road traffic policing and police officers without experience in road traffic policing.

## 3. Results

The results are presented in the following sequence: interrater reliability, reported pupil sizes, detection of the impairment of the driver, the role of work experience in traffic police on reported state of driver (impaired/not impaired), assessment of the impairment severity, and the role of work experience in traffic police on reported severity of the impairment.

### 3.1. Interrater Reliability of the VERIFY Checklist

Police officers who thought a driver was impaired completed the VERIFY checklist. The number of completed ratings for each video is listed in Table 4. Fleiss’ kappa revealed moderate to very good agreement between officers’ assessments (Table 4).

### 3.2. Reported Pupil Sizes

Figure 2 displays the estimated sizes of left and right pupils for each video, which were reported by officers who had undergone education in VERIFY. In seven cases, police officers rated the left and right pupil sizes differently.

### 3.3. Detection of Impairment of the Driver

Table 5 reports the results for the detection of impairment for each rated video separately. For each rated video, the results of the χ^2^ test are listed in the last column.

For the ratings on videos 1 and 2, χ^2^ revealed a significant difference between the three groups of participants. For the ratings collected on Video 1, the post hoc test displayed that the expected and observed frequencies significantly differed in the group of laypeople as well as in the group with VERIFY-trained police officers.

Laypeople were more likely to let an impaired driver continue driving than expected, whereas VERIFY-trained police officers were more likely to detect an impaired driver than expected. For the results collected in Video 2, the post hoc χ^2^ test revealed that the expected and observed frequencies significantly differed in the laypeople group. Laypeople were more likely to let the impaired driver continue driving than expected.

#### Role of Work Experience in Traffic Police on the Reported State of Driver (Impaired/Not Impaired)

When comparing the assessment of whether a driver was impaired or not between police officers with work experience and ratings of police officers without work experience in traffic police analogous, as done in the previous Section 3.3, no significant differences in ratings were found.

### 3.4. Assessment of Impairment Severity

Cronbach’s α analysis including the three components of the impairment severity scale (see Section 2.3) revealed a good reliability of the components (α = 0.84). Differences in the assessment of impairment severity between the groups are reported in Table 6.

The Bonferroni post hoc test of impairment severity rating in Video 1 revealed that the observed impairment severity was significantly different between all groups (*p* < 0.01). The estimated impairment severity increased from laypeople (M = 2.95, SD = 0.98) to the group without VERIFY education (M = 3.72, SD = 0.88) to the group with VERIFY-trained police officers (M = 4.06, SD = 0.88). The Bonferroni post hoc test of Video 2 revealed that laypeople (M = 2.77, SD = 0.73) differed significantly in their rating from the rating in the VERIFY-trained group (M = 3.42, SD = 0.79) and the group without training (M = 3.35, SD = 0.75, *p* < 0.01). The post hoc analysis of impairment severity rating in Video 3 revealed a significant difference (*p* < 0.01) between the group of trained police officers (M = 3.54, SD = 0.77) and laypeople (M = 2.97, SD = 0.80). There was also a significant difference in ratings between groups with and without training (M = 3.06, SD = 0.83).

#### Role of Work Experience in Traffic Police on the Reported Severity of Impairment

When comparing impairment severity rated by police officers with work experience in traffic police to ratings of police officers without work experience in traffic police analogous, as done in the previous Section 3.4, no significant differences in ratings were found.

## 4. Discussion

Our study aimed to validate an observational tool (VERIFY) for the detection of impairment of drivers presented on staged videos of a police traffic control. In particular, we evaluated the interrater reliability of the tool and how the perceived impairment level as assessed with the tool varied across laypeople, police officers without VERIFY education, and VERIFY-trained police officers. Additionally, we investigated whether the outcome when rating a driver’s impairment depended on whether police officers had experience in the traffic police force.

Our findings revealed a moderate to very good interrater reliability of the observational tool. High Fleiss’ kappa scores suggest that different evaluators observed behavior and appearance of the examinee similarly, indicating certain objectivity of the tool despite the subjective approach. However, a large variation in reported pupil sizes was observed (Figure 2) indicating that the estimation of pupil size using the size template alone is problematic. This finding is consistent with results reported by Olson et al. [31], who analyzed interrater reliability of subjective assessment of pupil size and reactivity. According to their results, the interrater reliability of both, size and reactivity estimation was low. Olson et al. [31] suggested considering automated pupillometry for a reliable assessment of pupil size and reactivity. An additional cause for the large variation in reported pupil sizes in the present study is that pupil sizes were reported from memory because the VERIFY checklist was filled in after the observation was completed. It is also likely that it is more difficult to estimate pupil size in a video setting with a short duration than in a real police traffic control, which takes more time than the duration of the video.

The results of the study indicate that identifying unimpaired drivers is easy, as there were no significant differences between the groups in assessing an unimpaired driver.

In one of the three videos showing an impaired driver, the rating difference between the groups was not significant (*p* = 0.08). The video showed a person under the influence of a stimulant. The person was very excited and had extremely large pupils. All groups predominantly suspected impairment, potentially because, in the general population, the effects of stimulant drugs that make a person hyperactive and cause large pupils are widely known. In the other two videos of impaired drivers, significant differences in impairment detection were observed. From these findings, we can postulate that laypeople are more likely to let a person who feels very ill and who might be under the influence of a medicinal drug continue driving (Video 2). This underestimation of diseases and the influence of medicinal drugs on driving behavior are consistent with previous studies [3,4]. Unfortunately, the use of objective screening devices to compensate for a lack of experience in laypeople is not feasible, as actual roadside screening devices cannot detect impairments caused by diseases or medicinal drugs.

In Video 1, a person who was sleepy and confused and who reacted very slowly was assessed as impaired more often by trained police officers than by police officers without training and by laypeople. In other words, trained police officers perceived the trait more clearly than the other two groups. The beneficial effect of training in reporting observed behavior was demonstrated in the study by Angkaw et al. [32], in which videotaped role-play interactions of participants with alcohol use disorders were rated. According to their results, observational training improves the interrater reliability and reduces the discrepancy in rating scores between the raters. Laypeople might not classify sleepy behavior as dangerous and therefore are more likely to let an impaired driver continue driving when clear indications of impairment are absent. The Swiss Council for Accident Prevention (BFU) presumes that the dark figure of accidents caused by sleepiness is as high as 10–20% [33], indicating that sleepy driving is strongly underestimated by the population. The effectiveness of the observational tool in identifying sleepiness is a major advantage over commonly used impairment detection methods.

While other widely used impairment tests, such as the drug recognition expert evaluation test, would have taken up to 90 min to form the suspicion of driving impairment, VERIFY enables detection of a driver’s impairment in a shorter time. From the present experience of the police of the canton of Zurich, VERIFY-trained police officers using the tool take approximately five minutes to form an opinion on a driver’s condition.

Our results also reveal a variation in the perception of the level of impairment across the three groups. In all videos of impaired drivers, laypeople perceived the impairment as less severe than VERIFY-trained police officers, and the differences between the two groups were significant. This finding is in line with the results reported by Rodriguez-Garzon et al. [34], who found that volunteer firefighters revealed a lower level of risk perception than professionals. On the other hand, trained police officers rated the level of impairment in unimpaired drivers lower than in laypeople, although the differences were not significant. Taken together, the differences in rating the level of impairment in impaired and unimpaired drivers suggest that trained police officers perform a more accurate rating compared to laypeople, which is also supported by the findings of Hahm et al. [20], who found that expert judgments are more accurate than others.

In contrast to the effect of training in applying the observational tool, officers with work experience in the traffic police force did not detect impairment significantly differently from police officers without experience in the traffic police force. This outcome suggests that skills in the use of the tool are acquired immediately or in a short time after officers receive the training.

### Limitations

This study has several limitations. For example, gender distribution differed between the groups. Most of the laypeople were female, whereas in the groups of police officers, the greater proportion was male. This should be kept in mind when interpreting the comparisons between the groups since a driver’s appearance or behavior could be perceived differently depending on gender. Hall et al. (2016) [35] reviewed research on gender differences in interpersonal accuracy and concluded that females excel at remembering others’ appearances and non-verbal behaviors, and they have more extensive knowledge of the meaning and use of non-verbal communication. This leads to the assumption that females tend to be better at assessing driving ability. However, training can improve accuracy of judgment of others’ emotions, personality traits, status, and intentions [36]. As police officers are trained to make such judgments and they need these skills in their daily work, it is assumed that the gender difference in the police groups is negligible. If the gender distribution in the laypeople group had been similar to that in the police groups, laypeople might have assessed driving ability less accurately than in the present study due to the smaller proportion of females. This suggests that if the gender distribution has been similar in the groups, the difference in the assessment of driving ability between the police officers and the laypeople would have been greater than in the present study.

Furthermore, 40% of the group without training and 69% with training had traffic police experience. The remaining proportion worked for the police but not as traffic police officers. One aim of the analysis was to test whether VERIFY training leads to a better result in assessing driving impairment than the group without VERIFY training. One could argue that the comparison is biased because the group without training included people who did not have experience in traffic policing. However, to reduce this bias, the χ^2^ and ANOVA tests were conducted with laypeople, police officers with experience in road traffic policing, and police officers without experience in road traffic policing. The results revealed no significant effect of experience on the expected frequency of rating the impairment correctly or assessing impairment severity. Furthermore, observation skills are required not only in the traffic police but also in many other areas of the police service. Therefore, it is very likely that the police officers of the group without training and with no experience as traffic police officers also need these skills in their daily work. It can be assumed that their skills are comparable to those of traffic police officers without any specific training in assessing driving impairment. Thus, this distributional inequality can be neglected.

One of the main limitations is that the traffic controls were presented in videos. Thus, police officers could not conduct the traffic controls themselves or interact with the drivers. Evaluating traffic control on the computer without interacting with the driver is not the same as conducting a real face-to-face control. The observer might recognize more evidence of driving impairment on the computer, as the attention only needs to be focused on the driver’s behavior and appearance and not on conducting the conversation. It could also be that raters would have assessed impairments better if they could conduct the control by themselves and interact directly with the driver. Based on the results of Hartwig et al. [24] and Ihlebæk et al. [37], it can be assumed that there would have been either no difference or a deterioration in the impairment assessment if the participants had conducted the road traffic control by themselves or observed it live.

Moreover, it could be argued that it would have to be proven that any staged behavior of impaired and unimpaired drivers represents what they really mean to define. The actors in the videos imitated the symptoms that often occurred from experience. Therefore, drivers’ behaviors in the videos are close to reality, and it can be assumed that the behaviors represent what they mean to define.

Furthermore, the interrater agreement and correct answer rate might depend on the obviousness of the acted behaviors. The results related to the videos of impaired drivers indicate that the interrater agreement might depend on the obviousness of the acted behaviors. The hyperactive person with large pupils shown in Video 3 is clearly impaired, and Cronbach’s α is higher than in the other two videos in which the signs of impairment are less apparent. However, the interrater reliabilities in the less apparent cases were still moderate. In terms of the correct answer rate, generally, the less apparent signs of impairment, the more difficult it is to identify an impaired driver. This is evident in the group of laypeople, whose correct answer rate is much higher in Video 3 than in the other two videos. However, there were no clear differences in the correct assessment between the three videos showing impaired drivers in the groups of police officers. This indicates that the correct answer rate does not necessarily depend on the obviousness of the acted behaviors.

Another limitation is that the videos did not allow participants to see all the details, such as the interior of the car, which could indicate a possible inability to drive (e.g., open alcohol bottles). However, the main goal of this study was to investigate whether the behavior and appearance were reported similarly by trained police officers, and this purpose was fulfilled in this study.

## 5. Conclusions

This study investigated the interrater reliability of an observational tool named VERIFY, which is used for detecting impaired drivers during road traffic controls. We assessed differences in rating driving impairment and its severity by laypeople, police officers trained to use the observational tool, and police officers without training. The results of our study revealed moderate to very good interrater reliability. The assessment of impaired drivers was significantly more reliable in police officers than in laypeople. Contrary to our expectations, trained police officers did not identify impaired drivers significantly better than police officers without training. However, trained police officers perceived the traits more clearly than those without training. This is particularly noticeable in cases where drivers do not display apparent impairment. In real-road traffic controls, drivers rarely display apparent impairments as in the videos used in this study. Thus, we assume that training brings an advantage over using the procedure without any training, especially in more difficult cases. With respect to the training in using the presented observational tool, our study indicates that training is successful, as trained police officers perceive driving impairment as more severe than police officers without training.

A combined application of objective screening methods for drug testing with the observational tool could help improve the reliability of drug testing through triangulation of the results recorded with the two methods. In addition, the tool could help reveal impairment by drugs for which no on-site objective drug testing devices are available.

Similarly as in traffic controls, the observational tool could also be applied to workplace drug testing (WDT) in general for coping with issues of an increasing demand for WDT. As listed by the US Substance Abuse and Mental Health Services Administration (SAMHSA), WDTs are applied in various occasions, such as pre-employment tests, for-cause and reasonable suspicion tests, and post-accident and post-treatment tests. Furthermore, the number of people taking drugs has been on the rise [38,39]. Therefore, there is a need for an accurate, effective, and quick method for identifying impaired workers. The observational tool presented in this study can help improve efficiency and effectiveness in workplace drug detection to cope with the increasing number of WDTs. The major point of criticism of the observational tool, namely subjectivity, could be mitigated in this study by demonstrating its good interrater reliability.

Our study supports the use of videos to familiarize police officers with the observational procedure. In the future, observational impairment evaluations could make use of video technology to remotely include experts trained in using the observational tool, thereby increasing the efficiency of testing.

Lastly, the findings in this study indicate that the population needs to be made more aware of the dangers of working under the influence of medicinal drugs or sleepiness.

In conclusion, the presented observational tool will improve efficiency and effectiveness in road traffic controls as well as in workplace drug testing and will therefore contribute to a safer work environment.

## Figures and Tables

**Figure 1 ijerph-19-07548-f001:**
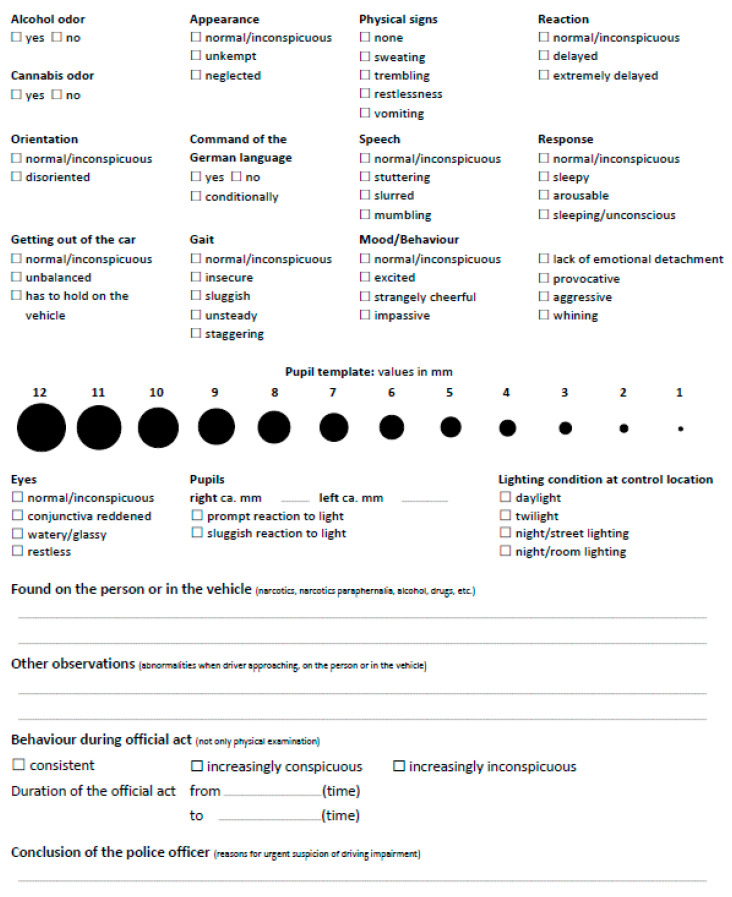
The VERIFY checklist consists of several items, which can be ticked if police officer thinks they apply to driver, and several open-ended questions. The VERIFY checklist is originally in German.

**Figure 2 ijerph-19-07548-f002:**
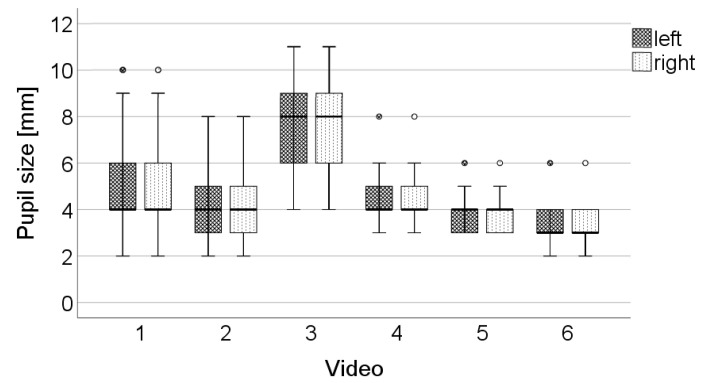
Boxplot representation of left and right pupil sizes per video. n: number of VERIFY-trained police officers who estimated pupil size. Videos 1–3 display impaired driver, and in videos 4–5, the driver is not impaired.

**Table 1 ijerph-19-07548-t001:** Sociodemographics and duration in minutes and seconds for completing the rating by groups.

		Group Laypeople	Without VERIFY Education	With VERIFY Education
**Age**	Min–Max	19–79	25–62	24–59
	M ^a^ (SD) ^b^	31.01 (13.14)	40.76 (11.22)	37.27 (7.71)
**Gender**	Female	54 (66.7%)	42 (28.8%)	31 (18%)
	Male	27 (33.3%)	104 (71.2%)	141 (82%)
	Total	81 (100%)	146 (100%)	172 (100%)
**Duration** **(min)**	Min	16:37	15:17	18:7
	Max	49.58	58:6	88:15
	M ^a^ (SD) ^b^	21:85 (5:47)	26:58 (7:54)	32:23 (10:77)

^a^ Mean ^b^ Standard deviation.

**Table 2 ijerph-19-07548-t002:** Self-reported years of experience of police officers in traffic police or in other police forces for police officers with and without education in the VERIFY procedure.

		Group without VERIFY Education	with VERIFY Education
**Experience in police force [years]**	M ^a^ (SD) ^b^	15.06 (11.82)	11.83 (7.56)
	N [total]	146	172
**Experience in traffic police force [years]**	M ^a^ (SD) ^b^	4.44 (5.24)	6.08 (5.27)
Currently works for traffic police	N	29	81
Worked for traffic police in the past	N	30	37
	N [total]	59	118

^a^ Mean ^b^ Standard deviation.

**Table 3 ijerph-19-07548-t003:** Details of videos showing staged traffic controls.

	Impairment of Stopped Driver	Signs of Impairment
Video 1	Under influence of alcohol or another depressant	Slow reaction; unfocused; confusion; imbalance; unsteady gait
Video 2	Feeling very ill, might be under influence of medicinal drug	Medicinal drug box in car; slow reaction; sleepy; unsteady gait
Video 3	Under influence of a stimulant	Hyper behavior; large pupils; inappropriately cheerful; lack of emotional detachment
Video 4	No impairment	
Video 5	No impairment	
Video 6	No impairment	

**Table 4 ijerph-19-07548-t004:** Interrater reliability of the checklist based on responses of officers who underwent education in VERIFY. Details on rated impairments are listed in Table 3.

Video	Impairment	n	Fleiss’ Kappa	*p*-Value	95% Cl
1	Impaired	146	0.48 ^c^	<0.01	[0.48, 0.48]
2	Impaired	142	0.48 ^c^	<0.01	[0.48, 0.48]
3	Impaired	154	0.73 ^b^	<0.01	[0.73, 0.73]
4	Unimpaired	9	0.90 ^a^	<0.01	[0.90, 0.91]
5	Unimpaired	18	0.75 ^b^	<0.01	[0.75, 0.75]
6	Unimpaired	16	0.67 ^b^	<0.01	[0.60, 0.60]

Cl = confidence interval, n = number of raters, ^a^ very good reliability, ^b^ good reliability, ^c^ moderate reliability according to Landis & Koch (1977).

**Table 5 ijerph-19-07548-t005:** Frequency analyses and χ^2^ tests for rating in driving impairment.

Video Driver’s Condition	Group	Correct Assessment	Incorrect Assessment	Total	χ^2^ Tests of Independence
Video 1 *impaired*	Laypeople	45 (55.6%)	36 (44.4%)	81(100%)	χ^2^(2) = 29.26
	withoutVerify	119 (81.5%)	27 (18.5%)	146 (100%)	*p* < 0.01
	withVerify	146 (84.9%)	26 (15.1%)	172 (100%)	φ = 0.27 ^a^
Video 2 *impaired*	Laypeople	52 (64.2%)	29 (35.8%)	81 (100%)	χ^2^(2) = 12.75
	withoutVerify	120 (82.2%)	26 (17.8%)	146 (100%)	*p* < 0.01
	withVerify	142 (82.6%)	30 (17.4%)	172 (100%)	φ = 0.18 ^a^
Video 3 *impaired*	Laypeople	67 (82.7%)	14 (17.3%)	81 (100%)	χ^2^(2) = 5.10
	withoutVerify	118 (80.8%)	28 (19.2%)	146 (100%)	*p* = 0.08
	withVerify	154 (89.5%)	18 (10.5%)	172 (100%)	φ = 0.11 ^a^
Video 4 *unimpaired*	Laypeople	79 (97.5%)	2 (2.5%)	81 (100%)	χ^2^(2) = 6.03
	withoutVerify	131 (89.7%)	15 (10.3%)	146 (100%)	*p* = 0.06
	withVerify	163 (94.8%)	9 (5.2%)	172 (100%)	φ = 0.12 ^a^
Video 5 *unimpaired*	Laypeople	72 (88.9%)	9 (11.1%)	81 (100%)	χ^2^(2) = 2.78
	withoutVerify	122 (83.6%)	24 (16.4%)	146 (100%)	*p* = 0.25
	withVerify	154 (89.5%)	18 (10.5%)	172 (100%)	φ = 0.08 ^a^
Video 6 *unimpaired*	Laypeople	72 (88.9%)	9 (11.1%)	81 (100%)	χ^2^(2) = 2.71
	withoutVerify	125 (85.6%)	21 (14.4%)	146 (100%)	*p* = 0.37
	withVerify	156 (90.7%)	16 (9.3%)	172 (100%)	φ = 0.07 ^a^

^a^ Small effect size [28]. φ = effect size (Cramer’s V).

**Table 6 ijerph-19-07548-t006:** Comparison of reported impairment severity between groups.

Video	Group	N	Impairment Score [Mean across All Participants and Variables II–IV (SD)]	One-Way ANOVA/Welch Test
Video 1 *impaired*	Laypeople	81	2.95 (0.98)	F(2, 396) = 41.98
	withoutVerify	146	3.72 (0.88)	*p* < 0.01
	withVerify	172	4.06 (0.88)	$n_{p}^{2}$= 0.18 ^a^
	Total	399	3.71 (0.99)	
Video 2 *impaired*	Laypeople	81	2.77 (0.73)	F(2, 396) = 21.31
	withoutVerify	146	3.35 (0.75)	*p* < 0.01
	withVerify	172	3.42 (0.79)	$n_{p}^{2}$ = 0.10 ^b^
	Total	399	3.26 (0.80)	
Video 3 *impaired*	Laypeople	81	2.97 (0.80)	F(2, 396) = 20.52
	withoutVerify	146	3.06 (0.83)	*p* < 0.01
	withVerify	172	3.54 (0.77)	$n_{p}^{2}$ = 0.10 ^b^
	Total	399	3.25 (0.84)	
Video 4 *unimpaired*	Laypeople	81	1.16 (0.37)	Welch’s F(2, 201.54) = 1.11
	withoutVerify	146	1.17 (0.54)	p = 0.33
	withVerify	172	1.11 (0.32)	$n_{p}^{2}$ = 0.005 ^c^
	Total	399	1.14 (0.42)	
Video 5 *unimpaired*	Laypeople	81	1.52 (0.69)	Welch’s F(2, 198.42) = 1.06
	withoutVerify	146	1.57 (0.68)	*p* = 0.35
	withVerify	172	1.48 (0.55)	$n_{p}^{2}$ = 0.005 ^c^
	Total	399	1.53 (0.63)	
Video 6 *unimpaired*	Laypeople	81	1.46 (0.58)	F(2, 396) = 0.20
	withoutVerify	146	1.41 (0.62)	*p* = 0.82
	withVerify	172	1.42 (0.67)	$n_{p}^{2}$ = 0.001 ^c^
	Total	399	1.42 (0.63)	

^a^ Large effect [28]. ^b^ Medium effect [28]. ^c^ Small effect [28]. $n_{p}^{2}$ = effect size (eta squared).

## Data Availability

The data presented in this study are available on request from the corresponding author.

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
