# Peer review of "Towards the Validation of an Observational Tool to Detect Impaired Drivers—An Online Video Study"

_ijerph, 2022, doi:10.3390/ijerph19127548_

Round 1
Reviewer 1 Report
The authors have done a terrific job in building the survey and presenting the results. However, the study needs to clearly outline the sampling technique to justify with references: why convenience sampling is appropriate? Also, I have ethical concerns as the author does not articulate how the data was anonymized and the ethical approval was obtained. Clarifying these aspects would make the article publication-ready.
Reviewer 2 Report
The present research performs a validation of an observation tool used to assess the impairment of drivers caused by alcohol. This is an interesting topic, with important practical applications in the field of traffic and road safety, considering the number of accidents caused by the human factor, especially by the consumption of alcohol and drugs at the wheel.
The introduction is adequate, as it explains the variables implied in the research. I would only recommend including the following references, in which transcultural research is explained, highlighting the impact of alcohol on the driving task through data from several countries (e,g, doi: 10.1186/1471-2458-10-205 and doi: 10.1186/1471-2458-11-526).
The methodology and results are adequately explained. However, I recommend complementing the discussion with more references that will contribute to the explanation of the obtained data. Moreover, I suggest including a detailed description of the practical implications offered by your research.
Reviewer 3 Report
The article tries to claim that video trained police officers are more capable of spotting impaired drivers than untrained personnel. The approach has several flaws.
The reviewer's main concerns are that
1) The study is biased. The "impaired persons" seem to be played by actors i.e. fake drivers (line 154ff). This is a serious problem as the acting impaired driver in the video might only partially resemble a real impaired person and hence, does not provide a sufficient statistics for the study. Hence, the results have very limited merit.
2) The study used ordinary persons, untrained and trained police staff. This is fine in gernal. Howver, the problem is that the individual groups exhibit different features and hence, are not comparable. Specificially, the laypersons "were chosen by the Department of Psychology at the University of Zurich" based on what criteria (besides the age)? Moreover, the trained staff had to answer a more exhaustive questionnaire than the other objects. Moreover, the ratio of male to female is unbalanced in the different groups.
3) Line 127 "The participants were excluded from the study when they took less than 20 minutes to complete the responses." what is the rational for choosing a threshold of 20min (not 19 o 21)? That sounds like random very random to the reviewer. How would the statistics change when the excluded participants were included in the study?
The authors base their claims on unpublished studies. Line 66-70 "In a forthcoming publication [17], we will ....". However, [17] does not exist, even not as accepted paper. This is bad practice. Hence, the claim and its reference has to be removed.
Language is very sloppy.
For example, line 31: "Alcohol....lead to productivity loss". This general prejustice is simply not supported by the provided reference [7] either. That all depends on the person, weight, age, psychology, percentage of alcohol (wine or spirituous beverages), etc.
Poor English language makes the text difficult to understand.
The right way of doing such a research is to consider two groups, a test group (trained) and a control group (untrained). Both groups must have THE SAME features (age, sex, education, experience, etc. ....). Then the test procedure must IDENTICAL for all objects. Only then, the conjecture made in the manuscript can be justified.
Minor comments:
Sometimes is is impossible to comprehend the text:
line 9: "Abuse of alcohol and drugs other than alcohol is a major"...... Abuse of alcohole other than alcohol does not make sense. Probably the phrase "other than alcohol" is to be removed.
line 23: "Observations could make use of virtually present trained police officers to improve efficiency ". This sentence is completely unclear. What kind of obersvations? "virtually present trained" Present or trained? The sentence has to be rephrased.
The same phrases appear over and over again. For example, line 76. ", police officers must undergo training in which observations are guided by experts." It has already been said before. Similarly in line 116.
Line 180: "After each video, participants in all groups were first asked whether they thought the 180
driver was unimpaired ". WHich driver? That in the video? if so, say it.
Round 2
Reviewer 3 Report
One minor issue. In Table 3, the videos are listed as 1,2,3,4,6,7 (note that #5 is missing). However, Tables 4-6 say 1,2,3,4,5,6 (note that #5 is present). There is an inconsistency in the numbering.
